# Development and Validation of the Breastfeeding Literacy Assessment Instrument (BLAI) for Obstetric Women

**DOI:** 10.3390/ijerph20053808

**Published:** 2023-02-21

**Authors:** María Jesús Valero-Chillerón, Rafael Vila-Candel, Desirée Mena-Tudela, Francisco Javier Soriano-Vidal, Víctor M. González-Chordá, Laura Andreu-Pejo, Aloma Antolí-Forner, Lledó Durán-García, Miryam Vicent-Ferrandis, María Eugenia Andrés-Alegre, Águeda Cervera-Gasch

**Affiliations:** 1Department of Nursing, Universitat Jaume I, Av de Vicent Sos Baynat, 12071 Castelló, Spain; 2Department of Nursing, Universitat de València, 46007 Valencia, Spain; 3Department of Obstetrics and Gynaecology, Hospital Universitario de la Ribera, 46600 Valencia, Spain; 4Foundation for the Promotion of Health and Biomedical Research in the Valencian Region (FISABIO-SP), 46020 Valencia, Spain; 5Department of Nursing, University of Alicante, 03080 Alicante, Spain; 6Department of Obstetrics and Gynaecology, Hospital Luis Alcanyis, 46800 Xàtiva, Spain; 7Nursing and Healthcare Research Unit (Investén-Isciii), Institute of Health Carlos III, 28029 Madrid, Spain; 8Department of Maternity, Hospital General Universitario, 12004 Castelló, Spain

**Keywords:** breastfeeding, breastfeeding literacy, questionnaires, validation study

## Abstract

Background: Despite international efforts to protect and promote exclusive breastfeeding (EBF) for infants up to six months of age, global rates of EBF continue to fall short of the targets proposed by the WHO for 2025. Previous studies have shown a relationship between the level of health literacy and the duration of EBF, although this relationship was not determinant, probably due to the use of a generic health literacy questionnaire. Therefore, this study aims to design and validate the first specific breastfeeding literacy instrument. Methods: A Breastfeeding Literacy instrument was developed. Content validation was carried out by a group of 10 experts in health literacy, breastfeeding or instrument validation, obtaining a Content Validity index in Scale (S-CVI/Ave) of 0.912. A multicentre cross-sectional study was carried out in three Spanish hospitals to determine the psychometric properties (construct validity and internal consistency). The questionnaire was administered to 204 women during the clinical puerperium. Results: The Kaiser-Meier-Oklin Test (KMO = 0.924) and Bartlett’s Test of Sphericity (*X*^2^ = 3119.861; *p* ≤ 0.001) confirmed the feasibility of the Exploratory Factor Analysis, which explained 60.54% of the variance with four factors. Conclusions: The Breastfeeding Literacy Assessment Instrument (BLAI) consisting of 26 items was validated.

## 1. Introduction

Pregnancy and the postpartum period constitute an important stage in women’s health, in which a series of events take place that require special attention and monitoring by the health system. Although it is a physiological process, it involves a continuum of decision-making in which women need to have sufficient information so that these decisions protect and promote not only their health, but also that of their children.

One of the most important decisions to be made is regarding the feeding the infant will receive. International organisations such as the World Health Organisation and UNICEF recommend exclusive breastfeeding (EBF) for the first six months of an infant’s life and breastfeeding with complementary foods until at least two years of age [1]. The promotion of EBF is an international target in different programmes such as the Comprehensive Implementation Plan on Maternal, Infant, and Young Child Nutrition of the World Health Assembly [2], the United Nations Decade of Action on Nutrition 2016–2025 [3], and the investment framework of the World Bank [4]. However, despite multiple efforts to protect breastfeeding (BF), rates of EBF at six months of infant life remain very low, at around 38% globally [5]. Furthermore, laws to protect breastfeeding remain inadequate in most countries [6]. In Europe, the six-month EBF rate is around 25% [7]. However, in Spain, the six-month EBF rate has varied from 16.8% in 2019 [8] to 39% in 2017 [9]. The data need to be interpreted with caution as the variation in these data is caused by the absence of a unified approach for collecting and monitoring BF information in Spain.

The premature discontinuation of breastfeeding is a complex phenomenon that is influenced by a multitude of factors, including demographic characteristics (e.g., young maternal age, low levels of education and socio-economic status), social considerations (e.g., inadequate workplace support), psychological determinants (e.g., maternal intentions before birth, self-assurance, and engagement in breastfeeding), as well as biological considerations (e.g., infant health concerns, maternal health issues, first-time motherhood, and issues related to lactation) [10,11,12,13]. These considerations contribute to the multifaceted nature of early breastfeeding cessation. However, several studies have shown that, in many cases, early weaning occurs due to maternal decisions or perceptions, which do not always correspond to reality [14]. In the face of these false perceptions, health literacy (HL) has a fundamental role because the primary outcome of having a good level of HL is the ability to make good decisions that promote and protect health [15].

Various authors have broadly defined the concept of HL over time [16]. Despite the lack of consensus on constructing a single definition of this concept, most authors agree that it is multidimensional, complex, and heterogeneous [17]. Sørensen et al. proposed an integrated model of HL that looked at cognitive and social skills that enable the individual to address four competencies (access, understand, appraise, and apply health information) and three domains in which the individual interacts with the health system (health care, disease prevention, and health promotion) [18].

This complex concept of HL has been reformulated and adapted to specific health areas or populations. As a result, it is possible to retrieve a multitude of validated instruments that allow us to generically assess the level of HL, such as the European Health Literacy Survey Questionnaire (HLS-EU-Q) [19] or the test of functional health literacy in adults (TOFHLA) [20]. There are also instruments available that focus on measuring literacy in specific health areas, such as the Literacy Assessment for Diabetes (LAD), which addresses diabetes literacy [21]. Others focus on specific populations, such as the eHealth Literacy Scale (eHEALS), which addresses electronic health literacy in a young population [22]. It is also possible to retrieve the Maternal Health Literacy Inventory in Pregnancy (MHELIP) instrument, which is designed to measure maternal health literacy [23]. However, to our knowledge, no previous instrument has measured breastfeeding literacy (BFL).

Recent studies have suggested that an adequate level of HL may be a protective factor against early BF cessation [12,13,24]. However, these studies use generic HL instruments to determine the relationship between HL and the specific health domain of BF. Specifically, they use the Short Assessment of Health Literacy for Spanish-speaking Adults (SAHLSA) [24] and the Newest Vital Sign (NVS) in its validated version for Spanish-speaking populations [12,13,24]. The main findings of using a generic instrument to explore a particular area of health lack specificity and concreteness in the results obtained, so the authors agree on the need for a specifically validated instrument to measure the level of BFL in women during the perinatal stage [12,13,24].

Therefore, this study aims to design and validate a specific instrument to measure the level of BFL.

## 2. Materials and Methods

### 2.1. Design, Setting, and Participants

A design and validation study of the Breastfeeding Literacy Assessment Instrument (BLAI) was conducted to assess the level of BFL in a Spanish context. The study took place from 1 December 2021 to 30 September 2022.

The project was designed under Organic Law 03/2018, of 5 December, under the Protection of Personal Data and Guarantee of Digital Rights. First, the instrument was designed by reviewing the literature and content validity by creating a panel of experts. Second, a cross-sectional study was carried out on women during the clinical postpartum period in three hospitals in the Valencian Community (Spain): Hospital Universitario de La Ribera (HULR); Hospital General de Castellón and Hospital Lluís Alcanyís de Xàtiva (Spain) to determine the psychometric properties of the BLAI. Inclusion criteria were: having given birth in one of the participating hospitals and voluntarily agreeing to participate in the study. Exclusion criteria were: having a linguistic barrier that impeded understanding and completion of the data collection form, multiple gestations, or the neonate being admitted to a neonatal care unit. Participants completed an online informed consent form prior to data collection. The Ethics and Research Committees of each participating hospital approved the study. Furthermore, the principles of the Declaration of Helsinki were respected throughout this effort.

According to Anthoine et al.’s recommendations for instrument validation, a sample size of between five and ten participants per instrument item is recommended [25]. Thus, given that the initial version of the instrument had 28 items, a sample of between 140 and 280 participants was required. However, according to Ferrando y Anguiano-Carrasco, a minimum sample size of 200 participants is recommended to assess the quality of a questionnaire [26]. Therefore, a sample size of at least 200 participants would be sufficient to satisfy both criteria. A non-probabilistic convenience sampling was performed, in which a data collection form was administered consisting of sociodemographic variables (age, country of origin, educational level, perceived socioeconomic status), obstetric variables (parity, feeding doubts before birth, previous BF, variables related to previous BF experience), and the BLAI.

### 2.2. Questionnaire Development and Content Validity

The BLAI was designed based on the definition of the HL concept adapted to the BF context. It was therefore organised into the following dimensions: D1: Access to breastfeeding-related information; D2: Understanding of such information; D3: Appraise the veracity of information related to breastfeeding; D4: Application of that information. The formulation of the items was based on the difficulty in dealing with the situations described, establishing a Likert-type scale with four response options to avoid central tendency errors. The items were developed based on the integrated model of health literacy proposed by Sørensen et al. [18]. This model considers the dimensions mentioned above and applies them to healthcare, disease prevention, and health promotion. Likewise, it considers the perspective of the individual’s capacity and the interaction that the individual has with the social and health environment. 

Following the development of the first battery of items, a panel of nine experts in breastfeeding, health literacy, and questionnaire development and validation, which included midwives, lactation consultants, and research nurses, was formed. The initial iteration of the survey instrument was presented to a panel of experts for an evaluation of its overall relevance, the appropriateness of individual items within the context of each dimension, and the identification of other item-specific feedback. As many rounds as necessary were carried out until an average congruence percentage (ACP) of 0.9, as recommended by the literature, was reached [27]. For this purpose, the Item Content Validity Index (I-CVI) was calculated using the methodology proposed by Polit and Beck, with considerations given to the level of validity of each item, the probability of agreement due to chance (Pc), and the modified Kappa coefficient [27]. In addition, the overall scale average (S-CVI) was calculated, which determines the mean of the scores of all the I-CVIs and reflects the overall validity of the instrument.

### 2.3. Psychometric Properties

After content validation, the instrument was administered to women in the participating hospitals during the clinical postpartum period, provided they voluntarily agreed to participate in the study. 

First, a descriptive analysis of the sample was carried out using the mean, standard deviation, and 95% confidence interval for quantitative variables and absolute and relative frequencies for qualitative variables. After this initial analysis, construct validity was studied using an exploratory factor analysis (EFA). For this purpose, the factor extraction method used was principal axis factorisation, applying an oblique factorial rotation, given the potential correlation between the different factors. The ProMax rotation method was used since a dominant factor was not considered. Previously, the feasibility of the EFA was confirmed with the Kaiser-Mayer-Olkin (KMO) test and Bartlett’s test of sphericity. A factor loading greater than 0.4 was considered to retain items in a given factor [28]. The dimensionality of the instrument was studied using the Kaiser criterion, which considers as many factors as eigenvalues greater than 1 are present [29].

Second, the instrument’s internal consistency and dimensions were determined. Since an ordinal response scale was used, McDonald’s Omega was employed (adequate internal consistency of ω = 0.7–0.9) [30]. Due to the non-normality of the overall scores for each dimension, Spearman’s correlation coefficient was used to investigate the relationship between the different elements of the instrument. A range between 0.50–0.70 was considered a good correlation, and >0.7 was a strong correlation [31].

### 2.4. Inferential Analysis

After studying the instrument’s psychometric properties, an inferential analysis was carried out to explore the association between the level of BFL and the rest of the variables included in the study, using Chi-squared or Fisher’s exact test, depending on the nature of the variables. Participants were first grouped by determining the cut-off points for each of the dimensions of the BLAI questionnaire using cluster analysis. The k-means method was used, forcing two groups to differentiate between inadequate and adequate BFL levels, obtaining statistically significant differences between the two groups.

Statistical analysis was carried out with SPSS v.26, considering a statistical significance level of *p* < 0.05.

## 3. Results

### 3.1. BLAI Validation Results

An ACP of 0.864 was achieved for content validity through the panel of experts (*n* = 9) after the first round. The experts’ contributions to reformulating some items were greatly valued; they added new items to cover certain aspects not contemplated and changed the dimension of others. After conducting a second round, the authors obtained an ACP score of 0.913, which met the percentage recommended by relevant research. After this second round, only minor modifications were made to the wording of the items, resulting in a version of the instrument consisting of 28 items (Access six items; Understand five items, Appraise ten items, Apply seven items). The wording of the items is available in the Appendix A, both in the original version in Spanish and in the translated version (not validated) in English.

Regarding the modelling of the instrument through exploratory factor analysis (EFA), it was observed that two items (Access6 and Appraise6) obtained a poor factor loading (<0.4) in the dimension for which they were developed. Moreover, according to theoretical reasoning, these two items had no place in another dimension. In addition, the instrument’s internal consistency slightly increased when these items were removed, so they were eliminated from the instrument, which went from 28 items to 26 items. 

Regarding the new 26-item version, KMO (0.924) and Bartlett’s Test of Sphericity (*X*^2^ = 3119.861; *p* ≤ 0.001) confirmed the feasibility of the EFA. The factor analysis explained 60.54% of the variance with a total of four factors, coinciding with the theoretical design of the instrument. Specifically, the first factor (Access) explained 44.02% of the variance and consisted of five items, the second factor (Apply) explained 8.04% of the variance and comprised seven items, the third factor (Appraise) explained 4.38% of the variance and consisted of nine items, and the fourth factor (Understand) explained 4.09% of the variance and consisted of five items. The overall reliability of the questionnaire (*ω* = 0.949) and of each of the dimensions (Access *ω* = 0.809; Understand *ω* = 0.810; Appraise *ω* = 0.912; Apply *ω* = 0.873) was excellent. Table 1 shows the results of the content validity, exploratory factor analysis, and reliability of the BLAI.

As also shown in Table 1, the structure matrix demonstrates that most items obtained a higher factor loading for the dimension they were designed for, except for the following seven items that showed a considerable factor loading for two different dimensions. The formulation of the Understand1 item does not fit into the Appraise dimension. The formulation of the Understand3 item could be considered in both the Access and Understand dimensions, although the theoretical reasoning gives it more weight in the Understand dimension. The wording of the Understand2 and Understand4 items means they do not fit into the Access dimension. Finally, Appraise1, Appraise2, and Apply3 cannot be included in the Understand dimension.

Regarding the correlation between the different dimensions, it is observed that all the correlations are good. Specifically, the correlation between the Appraise-Understand and Appraise-Apply dimensions is strong, as they are all statistically significant (Table 2).

Table 3 shows the minimum and maximum scores obtained in each dimension according to the cluster analysis carried out to differentiate between inadequate and adequate BFL. In addition, the descriptive analysis of BLAI for each of the dimensions can also be observed, in which it can be seen that the majority of the participants are in the category of Adequate BFL in all the dimensions, with the Understand dimension having the lowest percentage of women with Adequate BFL (54.9%, *n* = 112) and the Apply dimension having the highest percentage of women with Adequate BFL (66.2%, *n* = 135).

### 3.2. Descriptive Analysis

A total sample size of 204 participants was reached. The mean maternal age was 32.8 years (SD = 5.143; 95% CI 32.09–33.51). A total of 45.59% (*n* = 93) of the deliveries were attended at HULR, 83.8% (*n* = 171) of the women were originally from Spain, 50.5% (*n* = 103) had a university education, and 85.3% (*n* = 174) reported having a medium socioeconomic status (Table 4).

Regarding the type of breastfeeding at discharge, 74% (*n =* 151) of the women chose Exclusive Breastfeeding (EBF), 6.4% (*n* = 13) mixed breastfeeding, and 19.6% (*n* = 40) chose formula feeding. Table 5 shows variables related to the type of breastfeeding chosen during the puerperium. It was observed that 72.7% (*n* = 80) of primiparous women chose EBF. Of the women who opted for EBF, 82.3% (*n* = 135) had no doubts about the type of breastfeeding, while 38.5% (*n* = 15) did have doubts during gestation, although they finally chose EBF. Only one woman reported opting for EBF due to pressure from her environment.

As for the general perception of the previous BF experience (*n* = 82), 52.4% (*n* = 44) perceived it as a very good experience, and nine of them (10.7%) reported having a regular previous BF experience. Only 45.3% (*n* = 38) felt supported at all times by healthcare professionals, and 39.3% (*n* = 33) felt supported at all times by family and friends. The 63.4% (*n* = 52) fed EBF up to six months or more to their previous child. As for a reason for giving up breastfeeding, 36.9% (*n* = 31) of the cases were physiologically weaned, while 20.3% (*n* = 17) were weaned because they had started working.

### 3.3. Breastfeeding Literacy Assessment Instrument 

Table 6 shows that as the perceived socioeconomic level increases, the percentage of participants with adequate Access BFL increases (*p* = 0.016). It can also be seen that the percentage of women with adequate Understand BFL or adequate Apply BFL is higher in those women who offer EBF (Understand: 59.6%, *n* = 90, *p* = 0.023; Apply: 70.9%, *n* = 107, *p* = 0.026), while those who opted for mixed breastfeeding obtained a lower percentage (Understand: 23.01%, *n* = 3, *p* = 0.023; Apply: 38.5%, *n* = 5, *p* = 0.026). Regarding the Appraise dimension, the percentage of Adequate Appraise BFL is lower among primiparous women (*p* = 0.022), and the highest percentages are observed among multiparous women of second (78.9%, *n* = 56) or subsequent gestations (65.2%, *n* = 15), with the differences being statistically significant (*p* = 0.018). Regarding the Apply dimension, the percentage of women with Adequate Apply BFL is higher among multiparous women of second gestation (77.5%, *n* = 55), followed by primiparous women (60.9%; *n* = 67). Multiparous women of third or later gestations were the ones with the lowest percentage of Adequate Apply BFL. A comparative analysis of sociodemographic and BF-related variables for each of the dimensions of the BLAI questionnaire can be found in the Appendix A.

## 4. Discussion

The BLAI presents adequate psychometric properties to assess BFL levels in women during the perinatal period, with adequate construct validity and internal consistency. The exploratory factor analysis explains 60.54% of the variance with four domains, coinciding with the four dimensions covered by the concept of HL (Access, Understand, Appraise, and Apply) developed by Sørensen et al. [18]. 

It is worth mentioning that, during the instrument’s modelling, a number of items had a slightly higher loading in dimensions for which they were not designed. However, after thoroughly examining each item to evaluate the feasibility of assigning it to alternative dimensions, the research team determined that it was more appropriate to retain these items within their original dimensions, as the theoretical alignment was more convincing in these dimensions. In addition, two items were removed (Access6, Appraisse6) due to their poor factor loadings. The internal consistency of the BLAI slightly increased after their deletion.

As for the dimensionality study of the instrument, the EFA was run without determining a number of factors to extract, allowing the statistical programme to determine the number of factors based on the Kaiser criterion of eigenvalues greater than 1 [29]. This is the default method in the statistical programme used, and it is possible to retrieve scientific evidence that casts doubt on its practical usefulness, as has been reported by other authors [32,33]. However, the resulting factor structure coincided with the number of dimensions for which the instrument was created. Today, there are other, more commonly used methods to corroborate the appropriate number of factors, such as parallel analysis or the ratio of the first-to-second eigenvalue. However, we have not found a universally accepted criterion. For example, in the case of eigenvalues, there is no criterion for the ratio to be accepted, some authors propose four [34], others five [35], but none seem to be based on empirical reasoning. Therefore, it is essential that future studies consider other analyses for studying dimensionality.

While it is true that the use of a single criterion may lead to an overestimation or an underestimation of the actual number of factors, over-extraction leads to fewer measurement errors [36]. Moreover, it would not be appropriate to treat as unidimensional a construct of which the theoretical foundation is based on more than one factor, even if the multidimensionality is moderate. In the present instrument, an overall score of the construct would tend to lean towards the mean of the possible score range, and would not allow for discerning which competence/s the subject presents, and which others lower the mean score of the construct and would need to be addressed by a practitioner. Therefore, treating the construct in an unidimensional way would diminish its usefulness in practice. However, in order to obtain an instrument with a solid factor structure supported by theoretical and statistical reasoning, it is of utmost importance to progress with the validation process, with larger samples and different methods of studying dimensionality, in order to confirm or refute the factor structure that supports the theoretical reasoning.

In terms of the percentage of variance explained by each of the factors, it can be seen that the Access dimension is the one that explains the highest percentage of variance, followed by the Apply dimension. This may be because these dimensions are more manageable for women, while the Understand and Apply dimensions may be more complex due to the reflection involved in these situations. In other words, the general population can access information related to a given topic (Access) and apply the information they have accessed (Apply). However, people who are not experts in an area may find it more challenging to reflect on whether they adequately understand the information they have accessed (Understand), as well as to assess whether the source of information is reliable or may contain information that is not scientifically supported (Appraise). It is important that this finding is taken into account when addressing any health education, specifically in the area of BF, with the aim of training mothers-to-be, and even health professionals, to reflect on the information accessed in order to increase confidence when making health decisions based on the knowledge they have acquired. Future studies could address this necessary line of research.

As evidenced by the findings of this study, the BLAI questionnaire demonstrates utility in identifying areas where perinatal women may require additional competencies to access, understand, appraise and apply information about BF, not only for self-care purposes but also to prevent occurrences that may impede BF, as well as to foster successful initiation and continuation of BF. Similarly, it would be interesting in future studies to use the BLAI questionnaire to measure the effect of BF training or antenatal education on BF. Similarly, future studies should consider confirmatory factor analysis to confirm the current four-dimensional factor structure, as the evidence does not recommend using the same sample to address all validation phases of a newly created instrument, as this would lead to optimistic results [37]. In fact, we are currently continuing to collect data in order to be able to carry out the confirmatory factor analysis. However, this is the first publication derived from the design and validation of the instrument based on solid theoretical reasoning, so it is interesting to make its existence known, as well as its first psychometric properties.

This study is a continuation of previous studies that addressed the relationship between health literacy measured by generic instruments and BF [12,13,38]. It has not been possible to retrieve in the literature another validated instrument to address the level of BFL, which makes it challenging to contrast results in the present study. On the one hand, concerning age, the present study did not find a statistically significant association with the level of BFL, in line with the results of Vila-Candel et al., in which the study also showed no significant association with the level of LH [12]. On the other hand, Valero-Chillerón et al. did find that the mean age among mothers with an adequate level of BFL was higher than those with a limited level of BFL [13].

In the present study, no statistically significant association was observed between educational level and BFL level in any of the dimensions that comprise the questionnaire, in contrast to other studies that obtained such an association between HL level measured with generic instruments and educational level [12,13]. This may be due to the fact that two completely different phenomena; the level of education academically trains you in a certain area, whereas the level of breastfeeding literacy explores the individual’s ability to access information related to breastfeeding, understand that information, evaluate the quality of the information accessed, and apply that information in the specific area of breastfeeding. It is possible that a higher level of education may enhance an individual’s competence in certain areas of daily life, but it may not be sufficient to establish statistically significant relationships across all dimensions of the BFL concept. Another discrepancy is observed for parity. In the present study, a significant association was observed between Appraise BFL and the number of children; whereas this association was not significant in previous studies for HL levels [12,13]. In addition, Valero-Chillerón et al. observed an association between the country of origin and the level of HL, while this association could not be observed in the present study regarding the level of BFL, perhaps due to the low participation of women whose country of origin was not Spain [13]. In line with the findings of Sørensen et al., low socioeconomic status is related to low levels of HL, and, as in the present study, with Inadequate Access BFL [38].

It has not been possible to retrieve any study in which a statistically significant association was found between HL level measured with a generic instrument and maintenance of EBF at six months. However, Vila-Candel et al. did find a statistical association between LH level and maintenance of EBF at one, two, and four months of infant life, although they did not re-measure at six months [12]. Moreover, all studies seem to confirm the multi-causality derived from early breastfeeding cessation [12,13,24,39]. This is why it may not be appropriate to address this relationship using a generic instrument to give sufficient weight to the level of HL on the duration of EBF, and it may be advisable to use a specific instrument to assess the level of BFL. Future studies should address this aspect to confirm the results obtained.

It was observed that the percentage of women who opted for mixed breastfeeding had the lowest percentage of adequate understanding and adequate Apply BFL. This may be a chance finding due to the limited percentage of this category in the present study. Contrasting these results in future studies conducted with larger samples would be interesting. It is worth mentioning that the rates of EBF and mixed feeding are similar to those reported in the study by Chertok et al., and point to an increase in the numbers of mixed breastfeeding and formula feeding after the SARS-CoV-2 pandemic, due to the lack of support for breastfeeding during the pandemic, among other factors [40].

We must recognise several limitations in our study and cautiously interpret the results. Firstly, it should be noted that since we could not retrieve any previous instruments that measure the level of BFL or any other measurement method that could be used as a gold standard reference, it was not possible to study convergent validity. Secondly, it was challenging to randomise the study sample, so convenience sampling was used. Thirdly, it is necessary to advance the process of analysing the dimensionality of the instrument. The methods used in the present study need to be tested against more objective criteria in larger samples, minimising additional survey items to the BLAI questionnaire to try to avoid possible response bias among participants, in order to confirm the factor structure.

Despite the limitations, we believe that the good psychometric properties of the instrument suggest that its use should be considered, as it is the first validated instrument to measure the level of BFL. Previous studies have found that the percentage of women with limited HL was significantly higher among mothers who did not reach four months [12] or six months of EBF than among those who did reach EBF at these follow-up points [13,24]. Therefore, it is interesting to study the relationship between the level of BFL using the BLAI questionnaire and maintenance of EBF at six months, as well as to study the explanatory power of the instrument. Future studies will also allow us to contrast the results obtained and explore the possibility of refining the instrument or the suitability of maintaining the current version. Similarly, future studies could adapt and validate the current version of the instrument among health science professionals and students.

## 5. Conclusions

The Breastfeeding Literacy Assessment Instrument (BLAI) can be used as a valid questionnaire to assess women’s literacy during the perinatal period to access, understand, appraise, and apply information related to BF, both in the sphere of self-care and the prevention of problems that negatively impact on BF, as well as the promotion of the adequate establishment and maintenance of EBF. However, it would be interesting to use the BLAI questionnaire in future studies to corroborate its validity and reliability.

## Figures and Tables

**Table 1 ijerph-20-03808-t001:** Content validity, Exploratory Factor Analysis, and reliability of BLAI.

	Content Validity Index	Factors					Communalities
		1	2	3	4	*ω* ^1^	
Access	0.907					0.809	
Access1	1.00	0.676	0.399	0.311	0.402	0.948	0.490
Access2	1.00	0.748	0.403	0.419	0.401	0.948	0.560
Access3	1.00	0.687	0.318	0.356	0.411	0.948	0.505
Access4	1.00	0.656	0.367	0.391	0.552	0.948	0.563
Access5	0.67	0.565	0.405	0.445	0.557	0.947	0.534
Understand	0.956					0.810	
Understand1	1.00	0.426	0.460	0.551	0.529	0.947	0.476
Understand2	1.00	0.645	0.520	0.552	0.602	0.946	0.615
Understand3	0.89	0.713	0.422	0.473	0.615	0.947	0.601
Understand4	0.89	0.587	0.399	0.404	0.707	0.947	0.557
Understand5	1.00	0.494	0.485	0.480	0.725	0.947	0.526
Appraise	0.856					0.912	
Appraise1	0.89	0.510	0.601	0.673	0.741	0.946	0.715
Appraise2	0.78	0.508	0.580	0.634	0.679	0.946	0.654
Appraise3	0.78	0.340	0.665	0.696	0.676	0.946	0.608
Appraise4	1.00	0.338	0.639	0.703	0.603	0.946	0.584
Appraise5	0.67	0.330	0.513	0.771	0.396	0.947	0.589
Appraise7	1.00	0.394	0.549	0.801	0.525	0.947	0.654
Appraise8	0.89	0.423	0.610	0.749	0.605	0.946	0.616
Appraise9	0.89	0.489	0.590	0.692	0.616	0.946	0.625
Appraise10	0.78	0.371	0.708	0.729	0.555	0.946	0.653
Apply	0.968					0.873	
Apply1	1.00	0.380	0.706	0.618	0.645	0.946	0.611
Apply2	1.00	0.446	0.669	0.573	0.456	0.947	0.521
Apply3	1.00	0.471	0.587	0.472	0.607	0.947	0.481
Apply4	1.00	0.410	0.852	0.625	0.510	0.946	0.691
Apply5	1.00	0.485	0.776	0.587	0.494	0.946	0.618
Apply6	0.78	0.255	0.582	0.484	0.550	0.948	0.479
Apply7	1.00	0.335	0.692	0.482	0.537	0.947	0.534

**^1^** Internal Consistency measured with MacDonald’s Omega.

**Table 2 ijerph-20-03808-t002:** Correlation matrix between the dimensions of BLAI.

	Access	Understand	Appraise	Apply
Access	1.000			
Understand	0.680	1.000		
Appraise	0.546	0.707	1.000	
Apply	0.535	0.662	0.761	1.000

Rho de Spearman; All correlations are significant at the <0.001 level (bilateral).

**Table 3 ijerph-20-03808-t003:** Cut-off points between inadequate and adequate BFL and descriptive analysis of BLAI.

	Inadequate BFL				Adequate BFL				*p* ^3^
	Min	Max	*n* ^1^	% ^2^	Min	Max	*n*	%	
Access	1.8	3.00	82	40.2	3.20	4.00	122	59.8	<0.001
Understand	1.8	3.00	92	45.1	3.20	4.00	112	54.9	<0.001
Appraise	1.44	2.78	70	34.3	2.89	4.00	134	65.7	<0.001
Apply	1.29	2.71	69	33.8	2.86	4.00	135	66.2	<0.001

^1^ Absolute frequencies; ^2^ Relative frequencies; ^3^ Cluster analysis. BFL = breastfeeding literacy.

**Table 4 ijerph-20-03808-t004:** Sociodemographic Characteristics.

	*n* ^1^	% ^2^
Hospital		
H. Universitario de La Ribera	93	45.6
H. General Universitario de Castellón	88	43.1
H. Lluís Alcanyís de Xàtiva	23	11.3
Country of origin		
Spain	171	83.8
Central and South America	20	9.8
Rest of European Union Countries	9	4.4
Other	1	0.5
Educational level		
Primary studies	33	16.2
Professional training	68	33.3
Degree, bachelor’s degree	68	33.3
Master’s degree or Phd	35	17.2
Perceived socioeconomic status		
Low	26	12.7
Middle	174	85.3
High	4	2

^1^ Absolute frequencies; ^2^ Relative frequencies.

**Table 5 ijerph-20-03808-t005:** Descriptive analysis of obstetric and breastfeeding-related variables.

	Exclusive Breastfeeding		Mixed Feeding		Formula Feeding	
	*n* ^1^	% ^2^	*n*	%	*n*	%
Parity						
First	80	39.2	6	2.9	24	11.8
Second	55	27	5	2.5	11	5.4
Third or more	16	7.8	2	1.00	5	2.5
Feeding doubts before birth						
I had no doubts	135	66.2	6	2.9	23	11.3
I had doubts, but it was my own free will	15	7.4	7	3.4	17	8.3
I had doubts, I felt pressured	1	0.5	-	-	-	-
Previous BF						
Yes	70	34.3	5	2.5	7	3.4
No	81	39.7	8	3.9	31	15.2
General perception of previous breastfeeding experience (*n* = 82)						
Very good	43	51.2	1	1.2	-	-
Good	18	21.4	2	2.4	2	2.4
Regular	9	10.7	1	1.2	3	3.6
Bad	-	-	1	1.2	4	4.8
Professional support received during previous breastfeeding (*n* = 82)						
Supported at all times	33	39.3	2	2.4	3	3.6
Supported most of the times	12	14.3	2	2.4	2	2.4
Supported sometimes	8	9.5	1	1.2	1	1.2
Insufficient support	17	20.2	-	-	3	3.6
Support from family and friends received during previous breastfeeding (*n* = 82)						
Supported at all times	28	33.3	1	1.2	4	4.8
Supported most of the times	20	23.8	3	3.6	3	3.6
Supported sometimes	6	7.1	1	1.2	-	-
Insufficient support	16	19	-	-	2	2.4
Months exclusively breastfed during previous breastfeeding (*n* = 82)						
1 month or less	4	4.8	3	3.6	8	9.5
2–3 months	8	9.5	-	-	-	-
4–5 months	8	9.5	-	-	1	1.2
6 months or more	50	61	2	2.4	-	-
Main reason for abandonment of previous breastfeeding (*n* = 82)						
Previous BF has not ended	6	7.1	-	-	-	-
Physiological weaning	31	36.9	-	-	-	-
Breast problems unrelated to BF	1	1.2	-	-	2	2.4
Breast problems related to BF	2	2.4	-	-	2	2.4
Lack of professional support	1	1.2	1	1.2	-	-
Lack of family support	1	1.2	-	-	1	1.2
Work incorporation	15	17.9	2	2.4	-	-
Perceived lack of breastmilk	8	9.5	-	-	4	4.8
Reduced infant weight gain	5	6	2	2.4	-	-

^1^ Absolute frequencies; ^2^ Relative frequencies; BF: breastfeeding.

**Table 6 ijerph-20-03808-t006:** Statistically significant associations with the dimensions of the BLAI questionnaire.

	Inadequate BFL		Adequate BFL		*p*-Value
	*n* ^1^	% ^2^	*n*	%	
	Access				
Socioeconomic status					0.016 ^3^
Low	16	61.5	10	38.5	
Middle	66	37.9	108	62.1	
High	-	-	4	100	
	Understand				
Lactation type					0.023 ^4^
Exclusive Breastfeeding	61	40.4	90	59.6	
Mixed Feeding	10	76.9	3	23.1	
Formula feeding	21	52.5	19	47.5	
	Appraise				
Previous Breastfeeding					0.022 ^4^
Previous Breastfeeding	22	26.8	60	73.2	
No previous Breastfeeding	12	27.3	32	72.7	
Is my first pregnancy	35	46.1	41	53.9	
Parity					0.011 ^4^
First	47	42.7	63	57.3	
Second	15	21.1	56	78.9	
Third or more	8	34.8	15	65.2	
	Apply				
Parity					0.042 ^4^
First	43	39.1	67	60.9	
Second	16	22.5	55	77.5	
Third or more	10	43.5	13	56.5	
Lactation type					0.026 ^4^
Exclusive Breastfeeding	44	29.1	107	70.9	
Mixed Feeding	8	61.5	5	38.5	
Bottle feeding	17	42.5	23	57.5	

^1^ Absolute frequencies; ^2^ Relative frequencies; ^3^ Fisher’s exact test; ^4^ Chi-squared; BFL: breastfeeding literacy.

## Data Availability

Not applicable.

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
