# Peer review of "Development and Validation of the Breastfeeding Literacy Assessment Instrument (BLAI) for Obstetric Women"

_ijerph, 2023, doi:10.3390/ijerph20053808_

Round 1

Reviewer 1 Report

The authors present the development and validation of a breastfeeding literacy assessment instrument, the results of which could contribute to further study and strengthen intervention to improve breastfeeding practices in Spain and in the Spanish-speaking countries. 

Introduction. 

Line 52-53: Do the authors suggest that EBF rates are overestimated? Please, may you argue. 

Line 54-60: Given the high national and international mobilization, I suggest citing some study in migrant population (example: https://doi.org/10.3390/nu14153173). It can also be used in the discussion (line 326-327). 

 Methods.

What were the inclusion and exclusion criteria for participants?

Did participants give informed consent?

Discussion.

Line 318-322: These findings are very interesting, please elaborate further. What could be the reason for the non-association between educational level and BFL?

Others.

Please review all the bibliographical references again, i noticed some errors (example: ref. 13 and 24).

Author Response

Response to Reviewer 1 Comments

Thank you for your very positive and constructive feedback on our manuscript.

We have considered all your comments and suggestions (and the comments made by the other reviewer), attempting to improve/refine the original manuscript.

Below you will find a point-by-point response to your comments (in red).

The authors present the development and validation of a breastfeeding literacy assessment instrument, the results of which could contribute to further study and strengthen intervention to improve breastfeeding practices in Spain and in the Spanish-speaking countries.

Introduction.

Point 1: Line 52-53: Do the authors suggest that EBF rates are overestimated? Please, may you argue.

Response 1: Thank you for your highlight. We do not consider that the EBF rates are overestimated, but rather that the data should be interpreted with caution given that there is no unified criterion in Spain for collecting this data. Changes have been made on line 53

Point 2: Line 54-60: Given the high national and international mobilisation, I suggest citing some study in migrant population (example: https://doi.org/10.3390/nu14153173). It can also be used in the discussion (line 326-327).

Response 2: Thank you for your comment. However, we do not consider this a phenomenon that particularly affects the study population in our setting.

Methods.

Point 3: What were the inclusion and exclusion criteria for participants?

Response 3: Thank you for your suggestion. Changes have been made in lines 109-113.

Point 4: Did participants give informed consent?

Response 4: Thank you for your comment. An explanation has been made in lines 113 and 114.

Discussion.

Point 5: Line 318-322: These findings are very interesting, please elaborate further. What could be the reason for the non-association between educational level and BFL?

Response 5: Thank you for your highlight. In order to improve this suggestion, changes have been made in lines 365-372.

Others.

Point 6: Please review all the bibliographical references again, I noticed some errors (example: ref. 13 and 24).

Response 6: Thank you for your comment. Reference 24 has been deleted as a duplicate. The rest of the references have been reviewed; new references have been added to respond to other reviewers' comments. The pertinent changes have been made in the list of references and the citations in the text.

Reviewer 2 Report

This is an important study which validated a breastfeeding literacy assessment instrument. This is an important tool that could have large impacts. There are several improvements that should be made to improve the clarity of the results.

1. Methods, page 5 paragraph starting at 110. This paragraph is very confusing and I am unsure of the actual numbers of patients that were enrolled. This should be clear in this section. There is too much discussion of how that number was chosen. These citations would be good here, but the larger discourse should be moved to the discussion. As written, I am unsure if you met the criteria for both studies and where you ended up.

2. The abstract states that you developed a 28 item insturment; however, in the results it states that you removed 2 items to end up with 26 items. Please revise to make this clear.

3. Tables 4, 5, and 6 are very difficult to understand due to formatting. Please consider using left justification instead of centering and/or highlighting the titles. Also check where some start. Table 1 is much easier to read even while centered. For the lists you have left justification would make levels more apparent.

4. Paragraph starting at line 288 contains a wonderful description of what Access, Apply, and Appraise refer to along with the discussion. Consider moving the basics of how you are interpreting these variables to the methods section.

Author Response

Response to Reviewer 2 Comments

Thank you for your comments on our manuscript and for your constructive remarks.

We have considered all your comments and suggestions (and also the comments made by the other reviewer) attempting to improve/refine the original manuscript.

Below you will find a point-by-point response to your comments (in red).

This is an important study which validated a breastfeeding literacy assessment instrument. This is an important tool that could have large impacts. There are several improvements that should be made to improve the clarity of the results.

Point 1: Methods, page 5 paragraph starting at 110. This paragraph is very confusing and I am unsure of the actual numbers of patients that were enrolled. This should be clear in this section. There is too much discussion of how that number was chosen. These citations would be good here, but the larger discourse should be moved to the discussion. As written, I am unsure if you met the criteria for both studies and where you ended up.

Response 1: Thank you for your suggestions. Information was added in lines 122 and 123.

Point 2: The abstract states that you developed a 28 item insturment; however, in the results it states that you removed 2 items to end up with 26 items. Please revise to make this clear.

Response 2: Thank you for your comment. It has been amended in lines 23 and 31.

Point 3: Tables 4, 5, and 6 are very difficult to understand due to formatting. Please consider using left justification instead of centering and/or highlighting the titles. Also check where some start. Table 1 is much easier to read even while centered. For the lists you have left justification would make levels more apparent.

Response 3: Thank you very much for your comment. Accordingly, it has been modified in Tables 1, 4-6.

Point 4: Paragraph starting at line 288 contains a wonderful description of what Access, Apply, and Appraise refer to along with the discussion. Consider moving the basics of how you are interpreting these variables to the methods section.

Response 4: Thank you for your suggestion and feedback. Information has been added in lines 130-132.

Reviewer 3 Report

The manuscript covers an important topic, and the newly created measurement tool is immensely useful. The Breastfeeding Literacy Assessment Instrument (BLAI) has a thorough test creation process, and the introduction is well written and anchored.

My main criticism of the BLAI factor analysis methodology is as follows:

The hypothetical factor structure is not supported by the Exploratory Factor Analysis (EFA) factor weight matrix or the variances explained by the dimensions. Some objective criteria should be used to determine the number of factors: e.g. parallel analysis. 

The 4-factor structure is not supported by the factor weight matrix shown in Table 1, which instead suggests to a single-factor structure. High correlations between scales and the variances explained by each dimension, such as the ratio of first to second eigenvalues, prove this.

After the EFA, I believe it is crucial to conduct a Confirmatory Factor Analysis (CFA) to support the questionnaire structure because the factor structure is an important determinant of how well the instrument is evaluated and scored. I think it's necessary for the launch of this promising method.

Author Response

Response to Reviewer 3 Comments

Thank you for your comments on our manuscript and for your constructive remarks.

We have considered all your comments and suggestions (and also the comments made by the other reviewer) attempting to improve/refine the original manuscript.

Below you will find a point-by-point response to your comments (in red).

The manuscript covers an important topic, and the newly created measurement tool is immensely useful. The Breastfeeding Literacy Assessment Instrument (BLAI) has a thorough test creation process, and the introduction is well written and anchored.

My main criticism of the BLAI factor analysis methodology is as follows:

Point 1: The hypothetical factor structure is not supported by the Exploratory Factor Analysis (EFA) factor weight matrix or the variances explained by the dimensions. Some objective criteria should be used to determine the number of factors: e.g. parallel analysis.

Point 2: The 4-factor structure is not supported by the factor weight matrix shown in Table 1, which instead suggests to a single-factor structure. High correlations between scales and the variances explained by each dimension, such as the ratio of first to second eigenvalues, prove this.

Response to points 1 and 2: Thank you for your suggestion. In order to improve the comprehension of the paragraph, we have made changes related to the above two comments in lines 165-167 and 300-323.

Point 3: After the EFA, I believe it is crucial to conduct a Confirmatory Factor Analysis (CFA) to support the questionnaire structure because the factor structure is an important determinant of how well the instrument is evaluated and scored. I think it's necessary for the launch of this promising method.

Response 3: Thank you for your suggestion. We agree that a Confirmatory Factor Analysis (CFA) should be implemented; however, this is an exploratory study, and we have not conducted a CFA because it is not the aim of the study. We have clarified this point in lines 345-352. We thank you for your suggestion and are currently finalising the collection of a larger sample that will allow us to assess the BLAI scale.

Round 2

Reviewer 3 Report

Dear Authors,

Thanks for responding and considering my suggestions. My main aim is to help you generalise the results and to support your future work.

I think the questionnaire's psychometric explanations are imprecise. Dimensionality should be approached cautiously and leniently, in my opinion.

The empirical results tend to confirm the unidimensionality of the instrument, but this could be influenced by many things: e.g. 1. the sample size (n=204); 2. the response bias (if participants had to fill in a long questionnaire and/or if they were not sufficiently motivated, they did not differentiate much between the items' content; 3. in traditional factor analysis, we expect an item to be correlated with only one factor, which is often not a realistic expectation for psychological variables, which is why modern factor analyses include ESEM models that allow an item to be correlated with several dimensions. 4. The bifactor approach, which accepts general factor and single factors simultaneously, also appears to be a newer psychometric approach. This solution would, for example, allow the use of both the main scale score and subscales scores.

My detailed answers below are given in order to provide a more psychometrically accurate description of the arguments in favour of the statistical analyses carried out.    

Point 1: The hypothetical factor structure is not supported by the Exploratory Factor Analysis (EFA) factor weight matrix or the variances explained by the dimensions. Some objective criteria should be used to determine the number of factors: e.g. parallel analysis.

Point 2: The 4-factor structure is not supported by the factor weight matrix shown in Table 1, which instead suggests to a single-factor structure. High correlations between scales and the variances explained by each dimension, such as the ratio of first to second eigenvalues, prove this.

Response to points 1 and 2: Thank you for your suggestion. In order to improve the comprehension of the paragraph, we have made changes related to the above two comments in lines 165-167 and 300-323.

165-167: The dimensionality of the instrument was studied using the Kaiser criterion, which considers as many factors as eigenvalues greater than 1 are present [29].

I think the use and mention of the Kaiser criteria is not a good example. Recently, this traditional method has been very much criticized by experts in psychometrics.

See for instance (Watkins, 2018; Rogers, 2022): „Although selection of the correct number of factors to retain is one of the most important decisions in EFA (Child, 2006; Fabrigar & Wegener, 2012; Gorsuch, 1983; Izquierdo et al., 2014; Norman & Streiner, 2014), the default method used by many statistical software programs (e.g., the “eigenvalue 1” rule) is usually wrong and should not be used (Fabrigar & Wegener, 2012; Izquierdo et al., 2014; Norris & Lecavalier, 2010). Measurement specialists have conducted simulation studies and concluded that parallel analysis and MAP are the most accurate empirical estimates of the number of factors to retain and that scree is a useful subjective adjunct to the empirical estimates (Velicer, Eaton, & Fava, 2000; Velicer & Fava, 1998). Unfortunately, no method has been found to be correct in all situations (Fabrigar et al., 1999; Gorsuch, 1983; Pett et al., 2003), so it is necessary to employ multiple methods and carefully judge each plausible solution to identify the most appropriate factor solution (Fabrigar & Wegener, 2012; Gorsuch, 1983; Hair et al., 2010; Henson & Roberts, 2006; Izquierdo et al., 2014; Lloret et al., 2017; Loehlin & Beaujean, 2017; Norris & Lecavalier, 2010; Pett et al., 2003). Of course, relevant theory and prior research must also be included as evidential criteria (Gorsuch, 1983). Consequently, a range of plausible factor solutions should be evaluated by selecting the smallest and largest number of factors suggested by these multiple criteria.”

The traditional criteria for factor retention [Kaiser criterion (i.e., eigenvalue > 1), scree plot, and explained variance] have been consistently criticized for overestimating the number of factors (Gaskin & Happell, 2014; Howard, 2016; Izquierdo et al., 2014; Lloret et al., 2014). Any combination of those should be avoided (Lloret et al., 2014). The literature is harsh and emphatic on the use of the Kaiser criterion (the only objective classic criterion) to determine the number of factors. The Kaiser criterion was proposed half a century ago for reasons of computational efficiency. Recent simulation studies do not even consider the Kaiser criterion; they assume it is an inappropriate method and demonstrate that parallel analysis and the hull method perform better, particularly for ordinal data (Gaskin & Happell, 2014; Howard, 2016; Izquierdo et al., 2014; Lloret et al., 2014).”

303-323: As for the dimensionality study of the instrument, the EFA was run without determining a number of factors to extract and allowing the statistical programme to determine the number of factors based on the Kaiser criterion of eigenvalues greater than 1 [29]. The EFA resulted in a factor structure of four factors, coinciding with the number of dimensions for which it was created. It is true that there are other methods more commonly used nowadays to corroborate the appropriate number of factors, such as parallel analysis or the ratio of the first-to-second eigenvalue. However, in this case, parallel analysis is not suitable in instruments based on dichotomous or Likert-type information (as is the case of the present questionnaire which is approached using a Likert-type scale with four response options), since it is not appropriate to compare the eigenvalues generated by parallel analysis as it is based on random numbers with a normal distribution [33]. As for the ratio of the first-to-second eigenvalue, according to the evidence consulted, it can be determined that a ratio greater than four already determines the existence of unidimensionality [34], while other authors raise this ratio to 5 [35] and it is not possible to recover a consensus among authors on the specific ratio that should be adopted for this criterion. While it is true that the use of a single criterion may lead to overestimation or under-estimation of the actual number of factors, over-extraction leads to fewer measurement errors [36]. Moreover, it would not be appropriate to treat as unidimensional a construct whose theoretical foundation is based on more than one factor, even if the multidimensionality is moderate. In the present instrument, an overall score of the construct would tend to lean towards the mean of the possible score range, and would not allow discerning which competence/s the subject presents, and which others lower the mean score of the construct and would need to be addressed by a practitioner. Therefore, treating the construct in a unidimensional way would diminish its usefulness in practice.

However, in this case, parallel analysis is not suitable in instruments based on dichotomous or Likert-type information (as is the case of the present questionnaire which is approached using a Likert-type scale with four response options), since it is not appropriate to compare the eigenvalues generated by parallel analysis as it is based on random numbers with a normal distribution [33].

The above reasoning is not good, because then the factor analysis (EFA) could not have been carried out. Of course, it is very important to look at the distribution of our variables before choosing a factor analysis method. But this was not done here, there is no data on this in the manuscript. If distribution deviates significantly from the normal distribution, then there are several other possibility for analysis: e.g. using polycoric correlation. Parallel analysis is also well suited for dichotomous or Likert scales (see e.g. Weng & Cheng, 2005; Cho, & Bandalos, 2009; Timmerman & Lorenzo-Seva, 2011).

As for the ratio of the first-to-second eigenvalue, according to the evidence consulted, it can be determined that a ratio greater than four already determines the existence of unidimensionality [34], while other authors raise this ratio to 5 [35] and it is not possible to recover a consensus among authors on the specific ratio that should be adopted for this criterion.

It is of course true that this is not a clear-cut criterion, but I say this to indicate that there is no procedure that has been developed to determine the exact number of factors. This only confirmed my assumption.

Moreover, it would not be appropriate to treat as unidimensional a construct whose theoretical foundation is based on more than one factor, even if the multidimensionality is moderate. In the present instrument, an overall score of the construct would tend to lean towards the mean of the possible score range, and would not allow discerning which competence/s the subject presents, and which others lower the mean score of the construct and would need to be addressed by a practitioner. Therefore, treating the construct in a unidimensional way would diminish its usefulness in practice.

I fully accept the importance of a theoretical approach, but I want to ensure that psychometric results are not misrepresented and distorted.

So, I can accept that the 4-factor solution is considered the most professionally accepted by the instrument developers. However, this is not clearly confirmed by the factor weight matrix, and the justification for the statistics not being done is not professionally sound. 

Point 3: After the EFA, I believe it is crucial to conduct a Confirmatory Factor Analysis (CFA) to support the questionnaire structure because the factor structure is an important determinant of how well the instrument is evaluated and scored. I think it's necessary for the launch of this promising method.

Response 3: Thank you for your suggestion. We agree that a Confirmatory Factor Analysis (CFA) should be implemented; however, this is an exploratory study, and we have not conducted a CFA because it is not the aim of the study. We have clarified this point in lines 345-352. We thank you for your suggestion and are currently finalising the collection of a larger sample that will allow us to assess the BLAI scale.

345-352: Similarly, future studies should consider confirmatory factor analysis to confirm the current four-dimensional factor structure, as the evidence does not recommend using the same sample to address all validation phases of a newly created instrument, as this would lead to optimistic results [37]. In fact, we are currently continuing to collect data in order to be able to carry out the confirmatory factor analysis. However, this is the first publication derived from the design and validation of the instrument based on a solid theoretical reasoning, so it is interesting to make its existence known, as well as its first psychometric properties.

OK.  

References

Cho, S. J., Li, F., & Bandalos, D. (2009). Accuracy of the parallel analysis procedure with polychoric correlations. Educational and Psychological Measurement69(5), 748-759.

Rogers, P. (2022). Best practices for your exploratory factor analysis: A factor tutorial. Revista de Administração Contemporânea, 26.

Timmerman, M. E., & Lorenzo-Seva, U. (2011). Dimensionality assessment of ordered polytomous items with parallel analysis. Psychological Methods, 16(2), 209–220. https://doi.org/10.1037/a0023353

Watkins, M. W. (2018). Exploratory factor analysis: A guide to best practice. Journal of Black Psychology, 44(3), 219-246.

Weng, L. J., & Cheng, C. P. (2005). Parallel analysis with unidimensional binary data. Educational and Psychological Measurement, 65(5), 697-716.

Author Response

Dear Authors,

Thanks for responding and considering my suggestions. My main aim is to help you generalise the results and to support your future work.

First of all, we would like to convey our sincere thanks to you. Your valuable contributions not only raise the quality of the manuscript but have also enabled us to learn considerably. Below you will find a point-by-point response to your comments in round 2 (in orange).

Round 2, Point 1. I think the questionnaire's psychometric explanations are imprecise. Dimensionality should be approached cautiously and leniently, in my opinion.

The empirical results tend to confirm the unidimensionality of the instrument, but this could be influenced by many things: e.g. 1. the sample size (n=204); 2. the response bias (if participants had to fill in a long questionnaire and/or if they were not sufficiently motivated, they did not differentiate much between the items' content; 3. in traditional factor analysis, we expect an item to be correlated with only one factor, which is often not a realistic expectation for psychological variables, which is why modern factor analyses include ESEM models that allow an item to be correlated with several dimensions. 4. The bifactor approach, which accepts general factor and single factors simultaneously, also appears to be a newer psychometric approach. This solution would, for example, allow the use of both the main scale score and subscales scores.

Response to Point 1, Round 2: Thank you for your contributions. We add these limitations in lines 406-410.

My detailed answers below are given in order to provide a more psychometrically accurate description of the arguments in favour of the statistical analyses carried out.    

Point 1: The hypothetical factor structure is not supported by the Exploratory Factor Analysis (EFA) factor weight matrix or the variances explained by the dimensions. Some objective criteria should be used to determine the number of factors: e.g. parallel analysis.

Point 2: The 4-factor structure is not supported by the factor weight matrix shown in Table 1, which instead suggests to a single-factor structure. High correlations between scales and the variances explained by each dimension, such as the ratio of first to second eigenvalues, prove this.

Response to points 1 and 2: Thank you for your suggestion. In order to improve the comprehension of the paragraph, we have made changes related to the above two comments in lines 165-167 and 300-323.

165-167: The dimensionality of the instrument was studied using the Kaiser criterion, which considers as many factors as eigenvalues greater than 1 are present [29].

Round 2, Point 2: I think the use and mention of the Kaiser criteria is not a good example. Recently, this traditional method has been very much criticized by experts in psychometrics.

See for instance (Watkins, 2018; Rogers, 2022): „Although selection of the correct number of factors to retain is one of the most important decisions in EFA (Child, 2006; Fabrigar & Wegener, 2012; Gorsuch, 1983; Izquierdo et al., 2014; Norman & Streiner, 2014), the default method used by many statistical software programs (e.g., the “eigenvalue 1” rule) is usually wrong and should not be used (Fabrigar & Wegener, 2012; Izquierdo et al., 2014; Norris & Lecavalier, 2010). Measurement specialists have conducted simulation studies and concluded that parallel analysis and MAP are the most accurate empirical estimates of the number of factors to retain and that scree is a useful subjective adjunct to the empirical estimates (Velicer, Eaton, & Fava, 2000; Velicer & Fava, 1998). Unfortunately, no method has been found to be correct in all situations (Fabrigar et al., 1999; Gorsuch, 1983; Pett et al., 2003), so it is necessary to employ multiple methods and carefully judge each plausible solution to identify the most appropriate factor solution (Fabrigar & Wegener, 2012; Gorsuch, 1983; Hair et al., 2010; Henson & Roberts, 2006; Izquierdo et al., 2014; Lloret et al., 2017; Loehlin & Beaujean, 2017; Norris & Lecavalier, 2010; Pett et al., 2003). Of course, relevant theory and prior research must also be included as evidential criteria (Gorsuch, 1983). Consequently, a range of plausible factor solutions should be evaluated by selecting the smallest and largest number of factors suggested by these multiple criteria.”

The traditional criteria for factor retention [Kaiser criterion (i.e., eigenvalue > 1), scree plot, and explained variance] have been consistently criticized for overestimating the number of factors (Gaskin & Happell, 2014; Howard, 2016; Izquierdo et al., 2014; Lloret et al., 2014). Any combination of those should be avoided (Lloret et al., 2014). The literature is harsh and emphatic on the use of the Kaiser criterion (the only objective classic criterion) to determine the number of factors. The Kaiser criterion was proposed half a century ago for reasons of computational efficiency. Recent simulation studies do not even consider the Kaiser criterion; they assume it is an inappropriate method and demonstrate that parallel analysis and the hull method perform better, particularly for ordinal data (Gaskin & Happell, 2014; Howard, 2016; Izquierdo et al., 2014; Lloret et al., 2014).”

Response to Point 2, Round 2: Thank you for your comment, we have modified the lines where this criterion is mentioned in discussion (Lines 302-306).

303-323: As for the dimensionality study of the instrument, the EFA was run without determining a number of factors to extract and allowing the statistical programme to determine the number of factors based on the Kaiser criterion of eigenvalues greater than 1 [29]. The EFA resulted in a factor structure of four factors, coinciding with the number of dimensions for which it was created. It is true that there are other methods more commonly used nowadays to corroborate the appropriate number of factors, such as parallel analysis or the ratio of the first-to-second eigenvalue. However, in this case, parallel analysis is not suitable in instruments based on dichotomous or Likert-type information (as is the case of the present questionnaire which is approached using a Likert-type scale with four response options), since it is not appropriate to compare the eigenvalues generated by parallel analysis as it is based on random numbers with a normal distribution [33]. As for the ratio of the first-to-second eigenvalue, according to the evidence consulted, it can be determined that a ratio greater than four already determines the existence of unidimensionality [34], while other authors raise this ratio to 5 [35] and it is not possible to recover a consensus among authors on the specific ratio that should be adopted for this criterion. While it is true that the use of a single criterion may lead to overestimation or under-estimation of the actual number of factors, over-extraction leads to fewer measurement errors [36]. Moreover, it would not be appropriate to treat as unidimensional a construct whose theoretical foundation is based on more than one factor, even if the multidimensionality is moderate. In the present instrument, an overall score of the construct would tend to lean towards the mean of the possible score range, and would not allow discerning which competence/s the subject presents, and which others lower the mean score of the construct and would need to be addressed by a practitioner. Therefore, treating the construct in a unidimensional way would diminish its usefulness in practice.

However, in this case, parallel analysis is not suitable in instruments based on dichotomous or Likert-type information (as is the case of the present questionnaire which is approached using a Likert-type scale with four response options), since it is not appropriate to compare the eigenvalues generated by parallel analysis as it is based on random numbers with a normal distribution [33].

Round 2, Point 3: The above reasoning is not good, because then the factor analysis (EFA) could not have been carried out. Of course, it is very important to look at the distribution of our variables before choosing a factor analysis method. But this was not done here, there is no data on this in the manuscript. If distribution deviates significantly from the normal distribution, then there are several other possibility for analysis: e.g. using polycoric correlation. Parallel analysis is also well suited for dichotomous or Likert scales (see e.g. Weng & Cheng, 2005; Cho, & Bandalos, 2009; Timmerman & Lorenzo-Seva, 2011).

Response to Point 3, Round 2: Thank you for your contributions. We have redrafted this discussion paragraph. Specifically, the changes are on the lines 308-312.

As for the ratio of the first-to-second eigenvalue, according to the evidence consulted, it can be determined that a ratio greater than four already determines the existence of unidimensionality [34], while other authors raise this ratio to 5 [35] and it is not possible to recover a consensus among authors on the specific ratio that should be adopted for this criterion.

Round 2, Point 4. It is of course true that this is not a clear-cut criterion, but I say this to indicate that there is no procedure that has been developed to determine the exact number of factors. This only confirmed my assumption.

Response to Point 4, Round 2: Thank you for your contributions. In limitations we have added the weaknesses of the method used and the need to move forward with the study of dimensionality (Lines 406-410).

Moreover, it would not be appropriate to treat as unidimensional a construct whose theoretical foundation is based on more than one factor, even if the multidimensionality is moderate. In the present instrument, an overall score of the construct would tend to lean towards the mean of the possible score range, and would not allow discerning which competence/s the subject presents, and which others lower the mean score of the construct and would need to be addressed by a practitioner. Therefore, treating the construct in a unidimensional way would diminish its usefulness in practice.

Round 2, Point 5.I fully accept the importance of a theoretical approach, but I want to ensure that psychometric results are not misrepresented and distorted.

So, I can accept that the 4-factor solution is considered the most professionally accepted by the instrument developers. However, this is not clearly confirmed by the factor weight matrix, and the justification for the statistics not being done is not professionally sound. 

Response to Point 5, Round 2: Thank you for your comment, we have made changes to the lines 321-325.

Point 3: After the EFA, I believe it is crucial to conduct a Confirmatory Factor Analysis (CFA) to support the questionnaire structure because the factor structure is an important determinant of how well the instrument is evaluated and scored. I think it's necessary for the launch of this promising method.

Response 3: Thank you for your suggestion. We agree that a Confirmatory Factor Analysis (CFA) should be implemented; however, this is an exploratory study, and we have not conducted a CFA because it is not the aim of the study. We have clarified this point in lines 345-352. We thank you for your suggestion and are currently finalising the collection of a larger sample that will allow us to assess the BLAI scale.

345-352: Similarly, future studies should consider confirmatory factor analysis to confirm the current four-dimensional factor structure, as the evidence does not recommend using the same sample to address all validation phases of a newly created instrument, as this would lead to optimistic results [37]. In fact, we are currently continuing to collect data in order to be able to carry out the confirmatory factor analysis. However, this is the first publication derived from the design and validation of the instrument based on a solid theoretical reasoning, so it is interesting to make its existence known, as well as its first psychometric properties.

References

Cho, S. J., Li, F., & Bandalos, D. (2009). Accuracy of the parallel analysis procedure with polychoric correlations. Educational and Psychological Measurement69(5), 748-759.

Rogers, P. (2022). Best practices for your exploratory factor analysis: A factor tutorial. Revista de Administração Contemporânea, 26.

Timmerman, M. E., & Lorenzo-Seva, U. (2011). Dimensionality assessment of ordered polytomous items with parallel analysis. Psychological Methods, 16(2), 209–220. https://doi.org/10.1037/a0023353

Watkins, M. W. (2018). Exploratory factor analysis: A guide to best practice. Journal of Black Psychology, 44(3), 219-246.

Weng, L. J., & Cheng, C. P. (2005). Parallel analysis with unidimensional binary data. Educational and Psychological Measurement, 65(5), 697-716.
